# The Over-40-Years-Epidemic of Infectious Bursal Disease Virus in China

**DOI:** 10.3390/v14102253

**Published:** 2022-10-14

**Authors:** Wenying Zhang, Xiaomei Wang, Yulong Gao, Xiaole Qi

**Affiliations:** 1Avian Immunosuppressive Diseases Division, State Key Laboratory of Veterinary Biotechnology, Harbin Veterinary Research Institute, The Chinese Academy of Agricultural Sciences, Harbin 150069, China; 2World Organization for Animal Health (WOAH) Reference Laboratory for Infectious Bursal Disease, Harbin Veterinary Research Institute, The Chinese Academy of Agricultural Sciences, Harbin 150069, China

**Keywords:** infectious bursal disease virus, vvIBDV, nVarIBDV, epidemic, China

## Abstract

Infectious bursal disease (IBD) is an acute, highly contagious, immunosuppressive disease of chickens caused by the virus (IBDV), which critically threatens the development of the global chicken industry and causes huge economic losses. As a large country in the poultry industry, the epidemic history of IBDV in China for more than 40 years has been briefly discussed and summarized for the first time in this report. The first classic strain of IBDV appeared in China in the late 1970s. In the late 1980s and early 1990s, the very virulent IBDV (vvIBDV) rapidly swept across the entirety of China, threatening the healthy development of the poultry industry for more than 30 years. Variants of IBDV, after long-term latent circulation with the accumulation of mutations since the early 1990s, suddenly reappeared as novel variant strains (nVarIBDV) in China in the mid-2010s. Currently, there is a coexistence of various IBDV genotypes; the newly emerging nVarIBDV of A2dB1 and persistently circulating vvIBDV of A3B3 are the two predominant epidemic strains endangering the poultry industry. Continuous epidemiological testing and the development of new prevention and control agents are important and require more attention. This report is of great significance to scientific cognition and the comprehensive prevention and control of the IBDV epidemic.

## 1. Introduction

Infectious bursal disease (IBD) is an acute, highly contagious disease that affects chickens worldwide. It is caused by the infectious bursal disease virus (IBDV), which causes tissue damage and immunosuppression in infected flocks, making them susceptible to secondary or concurrent disease. The classic strain of IBDV infection first occurred in Gumboro, USA, in 1957 [1]. Approximately 30 years later, an antigenic variant strain of IBDV, evading the protection provided by the classic strain vaccine, was reported in Delaware, USA, in 1985 and is now predominantly epidemic in North America and Australia. Following this, the very virulent IBDV (vvIBDV) suddenly occurred in Belgium, Europe, in 1989 [2]. The vvIBDV is highly transmissible, and rapidly spread across Europe, Asia, Africa, and South America [3]. In China, IBD was first detected in Beijing and Guangdong in 1979 and rapidly spread to the main poultry areas of this country [4,5,6]. In recent years, the sudden prevalence of a novel variant of IBDV (nVarIBDV) in eastern Asia, including China, causing subclinical symptoms in chickens, has brought new threats and challenges to the poultry industry [7,8,9,10,11,12,13].

IBDV is a member of the genus *Avianbirnavirus*, a family of *Birnaviridae*, and is an icosahedral non-enveloped virus [14]. IBDV is a polyploid RNA virus with a genome consisting of two segments: A (3.2 kb) and B (2.7 kb). Segment A contains two partially overlapping open reading frames (ORFs): the smaller ORF in the front encodes the non-structural protein VP5, and the larger ORF in the back encodes the polyprotein, which is subsequently cleaved into a precursor protein pVP2, a viral serine protease VP4, and a capsid scaffolding protein VP3. Furthermore, pVP2 is processed into the mature capsid protein VP2 [15]. A hypervariable region (HVR, aa 206–350) of VP2 is responsible for the virulence, antigenic variation, and cell tropism of IBDV, playing a key role in its genetic evolution [16,17,18,19]. Segment B has only one ORF encoding the protein VP1, which has RNA-dependent RNA polymerase activity and is an essential protein for viral transcription and replication. Segment B and its VP1 also play an important role in the genetic evolution of IBDV and have a significant impact on its virulence [20,21,22,23].

As an RNA virus, the antigenic drift and escape caused by mutations pose great challenges to the prevention and control of IBDV. China has a large chicken industry, and there have been many clinical case reports of IBD in China. However, a systematic review of the overall epidemic characteristics of IBDV has not yet been reported. In this report, the genetic evolution history over the past 40 years and the current epidemic situation of IBDV in China have been systematically analyzed and summarized for the first time; this is of great significance to scientific cognition and the comprehensive prevention and control of IBDV epidemics.

## 2. The Coexistence of Various Strains of IBDV in China

Currently, circulating IBDVs include the classic (cIBDV), variant (varIBDV), very virulent (vvIBDV), and recently emerging novel variant strains (nVarIBDV). To understand the prevalence of IBDV in China, we collected IBDV nucleotide sequences from GenBank (https://www.ncbi.nlm.nih.gov/ (accessed on 31 December 2021)). As of 31 December 2021, there were 804 IBDV nucleotide sequences from China, including 520 *VP2*, 255 *VP1*, and 29 other genes. For genetic evolution analysis, excluding attenuated vaccine strains, 451 *VP2* nucleotide sequences covering a 435 bp fragment of *HVR* (bp 746–1180, aa 206–350), and 255 *VP1* nucleotide sequence, and covering a 430 bp fragment of B-marker (bp 441–867, aa 110–252) were selected. Traditionally, the *VP2 HVR* has often been used for genetic evolution analysis. Based on the *VP2* sequence analysis of 451 Chinese strains of IBDV, we found the highest number of vvIBDV (60.0%, 297/451), followed by nVarIBDV (30.1%, 136/451), with relatively few cIBDV (3.5%, 16/451) and varIBDV (0.4%, 2/451) (Figure 1a).

To clarify the current status of IBDV strains in China, we analyzed the spatiotemporal distribution of IBDV strains within various phenotypes in China (Figure 1). The epidemic of IBD can be traced back to the late 1970s [4,5], and the earliest report of IBDV in the classic strain CJ801 in China, with available gene sequences in GenBank, was published in 1980 [4,24]. Sequence analysis based on the *VP2* gene from GenBank (Figure 1) showed that varIBDV and vvIBDV appeared in China in the early 1990s, and that the *VP2* sequences of vvIBDV submitted to GenBank increased rapidly in the following 30 years. Since 2017, atypical IBD caused by nVarIBDV has become widespread in China, and nVarIBDV sequence numbers have shown a rapid growth trend, even exceeding the number of vvIBDV sequences (Figure 1b). Currently, the newly emerging nVarIBDV, and persistently circulating vvIBDV are the two predominant epidemic strains endangering the healthy development of the poultry industry, which is consistent with a recent clinical epidemiological survey from 2019 to 2020 [25]. The geographical distribution of IBDV in China was also analyzed. The isolation rates for IBDV in different regions were as follows: northwest (1%, 4/368), north (9%, 34/368), northeast (11%, 39/368), central (11%, 42/368), east (45%, 165/368), southwest (3%, 10/368), and South China (20%, 74/368). The IBDV number in East China (Anhui, Fujian, Jiangsu, Shandong, Shanghai, and Zhejiang provinces) was the highest (Figure 1c) because it is the most developed poultry industry in this region.

## 3. Genotype Classification of IBDV

Scientific classification is critical to correctly understanding the epidemic characteristics of viruses. Virus classification is not a one-step task but a process of gradual improvement. Traditionally, according to viral pathogenicity and antigenicity, serotype I strain of IBDVs have been divided into four phenotypes: classic [1], variant [26], very virulent [2], and attenuated IBDV. With the dazzling molecular characteristics of emerging strains owing to cumulative mutations, the traditional descriptive classification method can no longer meet the classification and definition of novel IBDV strains [27,28,29]. To solve this dilemma, Michel and Jackwood proposed a genogroup classification scheme in which serotype I strain of IBDVs were classified into seven genogroups according to *VP2* [30]. However, as a segmented virus, an IBDV classification scheme based only on a *VP2* coded by segment A is non-comprehensive.

Recently, a similar improved scheme for IBDV genotype classifications based on both segments was proposed [31,32], which is highly useful for the molecular epidemiology of IBDV. According to this scheme [31], based on the nucleotide sequences of the *VP2 HVR* and *VP1* B-markers, 86 representative strains isolated in China and 29 (*VP2*) or 13 (*VP1*) reference strains were used for phylogenetic analysis. The phylogenetic tree displayed that IBDVs were classified into nine genogroups of A (Figure 2a) and five genogroups of B (Figure 2b); the genogroup A2 was further divided into four lineages. Collectively, IBDVs circulating in China were classified as A1B1, A2dB1, A3B2/A3B3, and A8B1 genotypes corresponding to the classic, variant, very virulent, and attenuated IBDVs (Table 1), which showed the diversity of the epidemic strains in China.

## 4. The Persistently Circulating vvIBDV

Under the trend of a global vvIBDV pandemic in the late 1980s, the vvIBDV in China was first reported in the form of research articles in 1991 [6] and the first vvIBDV of the G9201 strain, with the gene sequence available in GenBank, was isolated in Guangdong province in 1992 [24]. Since then, with the rapid development of the chicken industry, vvIBDV has rapidly swept across almost the entire country of China (Figure 1). Owing to the high mortality and severe immunosuppression associated with this disease, vvIBDV has become one of the most important threats to the healthy development of the poultry industry in the past 30 years in China [24,25,27,33,34,35,36,37,38]. IBD is an important disease that must be immunized and prevented in all chicken farms in China.

Unlike other countries, it is highly interesting that there are two genotypes of vvIBDV, A3B2 and A3B3, as shown in Table 2. The genotype A3B2 of vvIBDV (typical vvIBDV) has been prevalent worldwide, including in China, since its first outbreak in Belgium in the late 1980s. Genotype A3B3 of vvIBDV (HLJ0504-like vvIBDV) is the most prevalent strain in China [34,39,40,41,42,43]. A high prevalence (86.0%) of A3B3-vvIBDV was reported in Southern China between 2000 and 2012 [34]. In a recent study of IBDV molecular epidemiology in China from 2019 to 2020, almost all detected vvIBDVs belonged to the genotype A3B3 [25]. Among the 86 vvIBDV strains with both *VP2* and *VP1* sequences analyzed in this study, genotypes A3B2 and A3B3 accounted for 34.9% (30/86) and 65.1% (56/86), respectively. Recently, it was reported that A3B3-vvIBDV has also been circulating in other countries, including Pakistan [44], India [45], Bangladesh [32,46], Thailand, Vietnam [30], Korea [9], and Venezuela [47]. Genotypes A3B2 and A3B3 of vvIBDV have the same segment A but different segment B of the genome. It has been speculated that segment B of the A3B3 strain originated from an unidentified ancestral virus circulating in avian or wild birds [40,48,49]. Although the homology of its segment B is between that of A3B2-vvIBDV and the attenuated strain, A3B3-vvIBDV has a very high mortality rate of over 60% in chickens as A3B2-vvIBDV. In recent years, vvIBDV infections have gradually been controlled due to the use of vaccines, as well as improvements in feeding and biosecurity management; however, the number of isolated vvIBDVs remain high, as illustrated in Figure 1B Therefore, the prevention and control of vvIBDV should be considered.

## 5. The Newly Emerging nVarIBDV

In recent years, farmers in China have consistently addressed the challenge of atypical IBD. This pathogen was identified as nVarIBDV for the first time in our laboratory [7]. The nVarIBDV belongs to genotype A2dB1, which is different from the early North American varIBDV (genotypes A2aB1, A2dB1, and A2dB1) [7,42]. Although varIBDV appeared in China in the early 1990s [24,50], most of their genetic information was not available. Sequence analysis showed that only two varIBDV strains, GZ902 and BX, had a high homology with the early North American varIBDV of genogroup A2a (Figure 2a). It has been reported that the variant IBDV spread from North America to China in the late 1980s and the early 1990s but did not cause a large-scale epidemic. After long-term latent circulation with the accumulation of amino acid mutations, it reappeared suddenly as nVarIBDV of the genotype A2dB1 in China in the mid-2010s [51].

Since 2017, nVarIBDV has been widely prevalent and has caused critical damage to major poultry breeding regions in China [7,10,52] and has also been reported in Japan [8], Korea [9,13], and Malaysia [11,12]. Atypical IBD caused by nVarIBDV pose a new threat to the poultry industry. It is well known that vvIBDV can cause the acute death of infected chickens with high mortalities within 3–5 days. Comparatively, nVarIBDV does not cause obvious appearance symptoms and death; however, the central immune organ bursa is severely destroyed causing severe immunosuppression in infected chickens and production performance is reduced. It was reported that nVarIBDV could suppress immune responses to vaccines against both highly pathogenic avian influenza [7] and Newcastle disease [53], the two most severe infectious diseases threatening poultry farming. In one study, the weight of nVarIBDV-infected broilers was dramatically reduced by approximately 16% compared to that of the control at 42 days of age, indicating huge economic losses [54]. Moreover, coinfection with nVarIBDV and other pathogens may further aggravate damage [55].

It was confirmed that nVarIBDV could partly circumvent the immune protection of existing vvIBDV vaccines, which is the key reason for the spread of nVarIBDV in immunized flocks [52,54]. Furthermore, residues 318 and 323 of the viral capsid protein are deeply involved in the antigenicity difference between the newly emerging nVarIBDV and persistently circulating vvIBDV [56]. Additional mechanisms of genetic variation and immune escape need to be further explored.

## 6. The Segment Reassortment and Gene Recombination among Circulating Strains

Segment reassortment among circulating strains is an important evolutionary characteristic that cannot be ignored for IBDV with a double-segmented genome. As shown in Figure 2 and Table 2, two types of segment-reassortment IBDVs between the vvIBDV and attenuated strains were observed among the Chinese strains. The genotype A3B1 of reassortment IBDV combines segment A from vvIBDV and segment B from the attenuated strain [57,58], and has also been reported in many countries, including Korea [59], India [60], Poland [61], Nigeria [62], Zambia [63], and Venezuela [47]. Another type of segment-reassortment IBDV (genotype A8B2) contains attenuated strains A and vvIBDV-B [64,65].

Most recently, segment reassortments of the two predominant epidemic strains, nVarIBDV and vvIBDV have also been observed. This reassortment strain (genotype A2dB3) has a nVarIBDV-A but an HLJ0504-like vvIBDV-B, which shows a similar pathogenicity to specific-pathogen-free (SPF) chickens as nVarIBDV [42]. However, another A2dB3 reassortment strain of YL160304 containing nVarIBDV-A and HLJ0504-like vvIBDV-B, isolated in southern China, could enhance its pathogenicity with 10% mortality compared to that of nVarIBDV [43].

In addition to segment reassortment, gene recombination may play an important role in the evolution of IBDV. Recently, a homologous recombination between nVarIBDV and an intermediate vaccine strain was identified, in which the 3′ regions of segment A in nVarIBDV were replaced by the corresponding region of an intermediate vaccine strain. Compared with the pathogenicity of nVarIBDV, this recombinant strain showed an increased viral pathogenicity in chick embryos [66]. Segment recombination and gene reassortment among the circulating strains increased the epidemiological complexity, potential harm and raised the difficulty of preventing and controlling IBDV.

## 7. Immune Prevention and Control of IBDV

Vaccines are an effective method for preventing and controlling vvIBDV infections. In China, the vvIBDV vaccine is a necessary routine vaccine, especially for large-scale intensive chicken farms. However, the vvIBDV vaccines currently available in China also require improvement. First, there are obvious biological safety risks due to the widespread use of intermediate and hot vaccine strains [67]. Second, most of the vaccine strains in use are non-domestic strains with possible incomplete antigenicity matches.

Additionally, the sudden prevalence of nVarIBDV poses a great challenge for the comprehensive control of IBD. To some extent, nVarIBDV can escape the immune protection of existing vvIBDV vaccines because the vvIBDV-antiserum cannot neutralize nVarIBDV well; therefore, there is an urgent need to develop a novel vaccine that completely matches the antigenicity of nVarIBDV [52,54,56]. It was reported that a tailored vaccine based on reverse genetics could provide 100% protection against newly emerging nVarIBDV and the persistently circulating vvIBDV [68]. Recently, viral-like particle (VLP) candidate vaccines have also been developed [69,70], one of which could provide complete immune protection against the two predominant epidemic strains of the homologous nVarIBDV and the heterologous vvIBDV [69].

For modern intensive poultry-farming, to some extent, immunity is productivity. In addition to vaccines, there are several other factors that are very important for improving the immunity of poultry flocks. First, biosafety: strict biosafety measures will greatly reduce the prevalence of epidemic diseases and secondary infections. The second is feeding management. Good nutrition ratios and appropriate temperatures, moderation, and ventilation can improve the resistance of chickens to disease and reduce conditional disease. Third, animal welfare: reducing resistance, replacing resistance, and reasonably adjusting breeding densities are a green breeding concept that must be adhered to.

In conclusion, according to the literature and sequence analysis, the epidemic history of IBDV in China for more than 40 years has been briefly reviewed and summarized for the first time. IBDV of the classic strain appeared in China in the late 1970s. In the late 1980s and early 1990s, vvIBDV rapidly swept through China, threatening the healthy development of the poultry industry for more than 30 years. The variant IBDV, after long-term latent circulation with the accumulation of mutations, reappeared suddenly as nVarIBDV in China in the mid-2010s. Currently, there is a coexistence of various IBDV genotypes; the newly emerging nVarIBDV of A2dB1 and the persistently circulating vvIBDV of A3B3 are the two predominant epidemic strains endangering the poultry industry. Continuous epidemiological testing and the development of new prevention and control agents are important and require more attention.

## Figures and Tables

**Figure 1 viruses-14-02253-f001:**
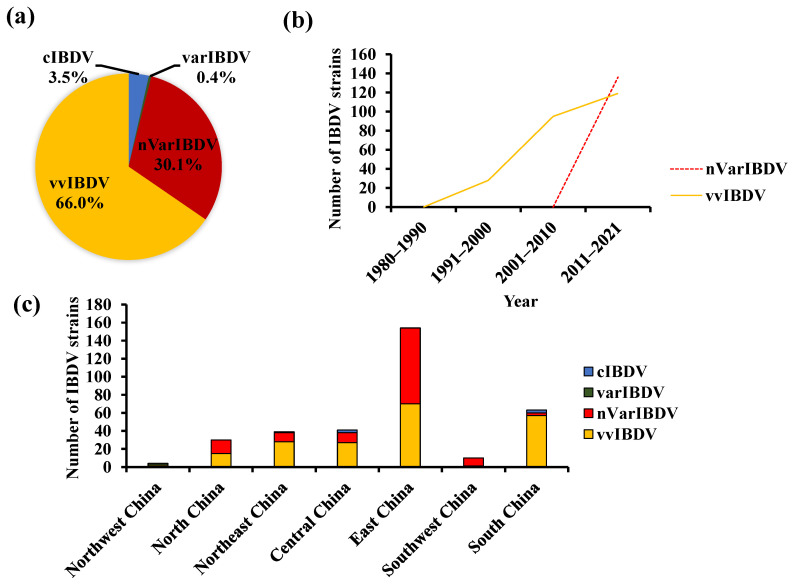
Spatiotemporal analysis of infectious bursal disease virus (IBDV) strains with various phenotypes in China. (**a**) The proportion of IBDV strains with various phenotypes. (**b**) Temporal distribution. (**c**) Geographical distribution.

**Figure 2 viruses-14-02253-f002:**
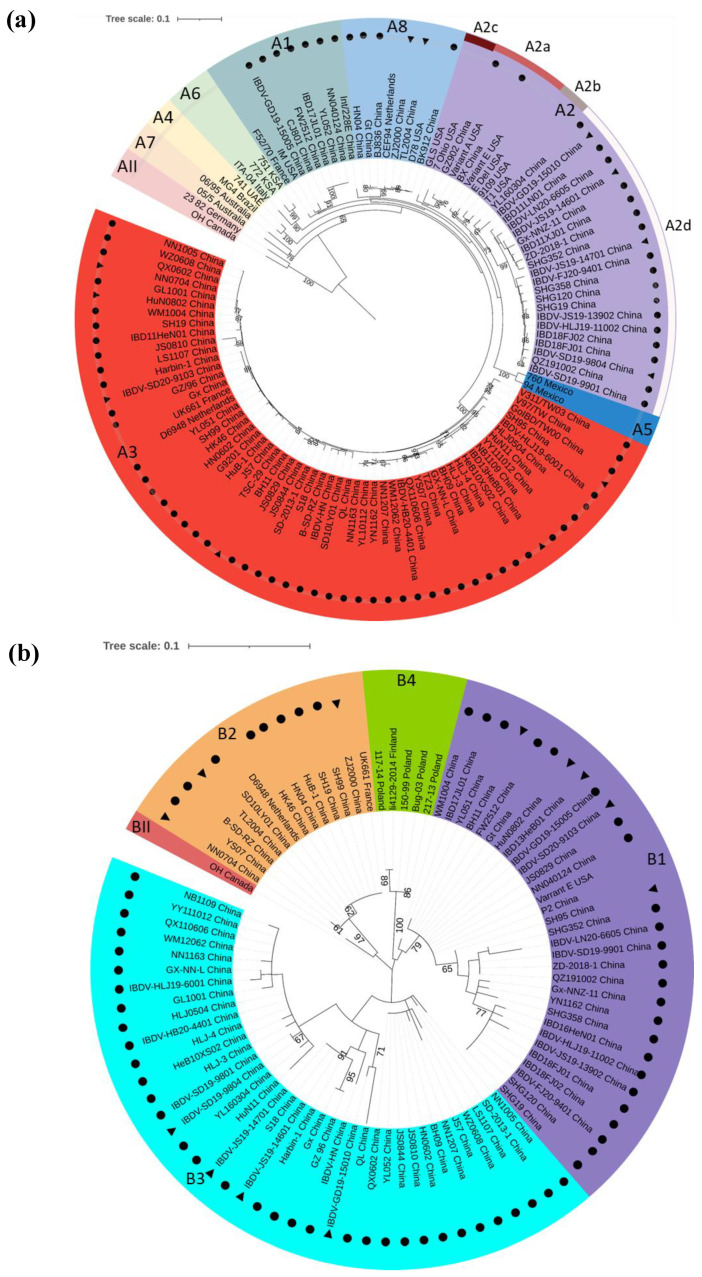
Phylogenetic analysis of the nucleotide sequences of the representative strains of infectious bursal disease virus (IBDV) in China. (**a**) Nine genogroups based on the hypervariable region (HVR) of *VP2* gene in segment A. (**b**) Five genogroups based on the B-marker of the *VP1* gene in segment B. The trees are generated by the maximum-likelihood method using MEGA7 software and are visualized using iTOL. The trees are drawn to scale with branch lengths measured in the number of substitutions per site. Only branches supported by a bootstrap value above 60% are displayed. A total of 86 representative strains isolated in China and 29 (*VP2*) or 13 (*VP1*) reference strains are involved in this analysis. All Chinese strains are highlighted with a solid circle (●), of which 13 stains with segment-reassortant characteristics are marked with a solid triangle (▲).

**Table 1 viruses-14-02253-t001:** The genotype classification corresponding to the traditional phenotype of IBDV.

Phenotype	Genotype	Reference Strain	GenBank No. (A, B)
Classic strains	A1B1	YL052 China	DQ656521, KC968889
		FW2512 China	DQ656499, KC986359
		NN040124 China	DQ656502, KC968872
		IBD17JL01 China	MN604241, MN604242
		IBDV-GD19-15005 China	MW682890, MW863620
Variant strains	A2aB1	Variant E USA	AF133904, AF133905
	A2bB1	9109 USA	AY462027, AY459321
	A2cB1	GLS USA	AY368653, AY368654
Novel variant strains	A2dB1	SHG19 China	MH879045, MH879092
		SHG120 China	MH879063, MH879110
		ZD-2018-1 China	MN485882, MN485883
		Gx-NNZ-11 China	JX134483, JX134484
		QZ191002 China	MZ066613, MZ066615
vvIBDV	A3B2	NN0704 China	FJ615511, KC968858
		YS07 China	FJ695138, FJ695139
		B-SD-RZ China	GQ166972, GQ166971
		SD10LY01 China	KF569803, KF569804
		HuB-1 China	KF569805, KF569804
	A3B3	Gx China	AY444873, AY705393
		S18 China	MK472711, MK472712
		QL China	JX682709, JX682710
		GL1001 China	KC968831, HQ452814
		HLJ0504 China	GQ166972, GQ451331
Attenuated strains	A8B1	Gt China	DQ403248, DQ403249
		BH15 China	DQ656498, KC968825
		JD1 China	AF321055, AY103464
		HuN0804 China	FJ615498, KC968842
		QX110603 China	KC918849, KC968876

**Table 2 viruses-14-02253-t002:** Segment reassortment strains of IBDV.

Genotype	Feature ^1^	Reference Strain	GenBank No. (A, B)
A3B1	vv-A/att-B	SH95 China	AY134874, AY134875
77.5% (31/40)		TSC-2(9) China	DQ656519, KC968881
		BH11 China	DQ656497, KC968823
		NB1109 China	KC918838, KC968855
		JS0829 China	FJ615508, KC968853
		JS0822 China	FJ615507, KC968852
		JS0821 China	FJ615506, KC968851
		JS0819 China	FJ615505, KC968850
		JS0811 China	FJ615504, KC968849
		JS0809 China	FJ615502, KC968847
		GL0902 China	HQ452817, KC968830
		GL0901 China	HQ452816, KC968829
		NN0603 China	FJ615509, KC968856
		JS0806 China	FJ615501, KC968856
		HuN0802 China	FJ615496, KC968840
		HuN0801 China	FJ615495, KC968839
		WM1004 China	JQ260883, KC968883
		YN1162 China	JQ260876, KC968894
		YN1161 China	JQ260875, KC968894
		NN1166 China	JQ260874, KC968866
		NN1165 China	JQ260873, KC968865
		NN1164 China	JQ260872, KC968864
		YL051 China	DQ656506, KC968888
		IBDV-HN China	KT884486, KY948019
		IBDV-SD20-9103 China	MZ766382, MZ766407
		IBDV-SD20-9104 China	MZ766383, MZ766408
		IBDV-SD20-9102 China	MZ766381, MZ766406
		IBD13HeB01 China	KP676467, KP676468
		NN0704 China	FJ615511, KC968858
		NN1007 China	JQ260882, KC968860
		NN1005 China	JQ260881, KC968859
A2dB3	nVar-A/uniq-B	IBDV-JS19-14701 China	MW700332, MW700333
12.5% (5/40)		IBDV-JS19-14601 China	MW682905, MW863641
		IBDV-GD1915010 China	MW682891, MW863621
		IBDV-SD19-9801 China	MW682907, MW863642
		IBDV-SD19-9804 China	MW682908, MW863643
A8B2	att-A/vv-B	ZJ2000 China	AF321056, DQ166818
10% (4/40)		TL2004 China	DQ088175, DQ118374
		HN04 China	KC109816, KC109815
		YL160304 China	MZ066614, MZ066616

^1^ vv—very virulent IBDV; nVar—novel variant IBDV; att—attenuated strain; uniq—unique strain as HLJ0504-like IBDV.

## Data Availability

Data can be requested by writing to the author.

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
