# Peer review of "The Over-40-Years-Epidemic of Infectious Bursal Disease Virus in China"

_viruses, 2022, doi:10.3390/v14102253_

Round 1

Reviewer 1 Report

The manuscipt is well designed and will contribute sugnifanctly to poultry production. Although the authors coverd all topics related to IBDV, there a one part may add values to this great review. In my opinion, it will be great to add a paragraph discuss the nutritional startiges that may mitigate the negative effects of IBVD. In addition, management and welfare startgies in the farm to mitigate the effcts of IBVD in China or Asian countries.

Reviewer 2 Report

Reviewer comments:

Title: The Over-40-Years-Epidemic of Infectious Bursal Disease Virus 2 in China

Authors: Zhang et al.,

General Comments:

Infectious bursal disease virus (IBDV) causes an acute, immunosuppressive disease of chickens and economically important worldwide. After its first appearance in 1970s in China, IBDV evolved so quickly and vvIDV and nVarIBDV have emerged. Monitoring and investigation of circulating strains and variants will help to establish new vaccination, preventive and control strategies.This is an important review that gives information on the situation of IBDV in China in last 40 years. Also, authors of this review has previously published studies on IBDV.

Abstract:

Well written and summarises the content if the review.

Introduction:

Well written and reflects the content of the review.

Genotype classification (Line 89)

For readers working outside China and there is big trade with China, it will be better to include vvIBDVs and nvarIBDVs to the existing trees  from all neighboring countries and Iran, Turkey and Russia. 

Immune prevention and control (Line 219)

The virus neutralisation data of the vvIBDV and nvarIBDv needs to be included. Also, if there is vaccine escape mutants they should be mentioned here. 

Fan et al., 2019 and other data.

Authors indicate that the farmers are seeing atypical IBDV infections in chickens in China, it woud be better to include, if there is, differential clinical signs and necropsy findings related to vvIBDv and nvarIBDV. 
